# CURRICULUM LEARNING FOR DEEP GENERATIVE MODELS WITH CLUSTERING

## ABSTRACT

Training generative models like Generative Adversarial Networks (GANs) is challenging for noisy data. A novel curriculum learning algorithm pertaining to clustering is proposed to address this issue in this paper. The curriculum construction is based on the centrality of underlying clusters in data points. The data points of high centrality takes the priority of being fed into generative models during training. To make our algorithm scalable to large-scale data, the active set is devised, in the sense that every round of training proceeds only on an active subset containing a small fraction of already trained data and the incremental data of lower centrality. Moreover, the geometric analysis is presented to interpret the necessity of cluster curriculum for generative models. The experiments on the cat and human-face data validate that our algorithm is able to learn the optimal generative models (e.g. ProGAN) with respect to specified quality metrics for noisy data. An interesting finding is that the optimal cluster curriculum is closely related to the critical point of the geometric percolation process formulated in the paper. Besides, the centrality-accuracy curve in our algorithm provides an effective way of visualizing the model state with curriculum training.

## 1 INTRODUCTION

Deep generative models have piqued researchers' interest in the past decade. The fruitful progress has been achieved on this topic, such as auto-encoder (Hinton & Salakhutdinov, 2006) and variational auto-encoder (VAE) (Kingma & Welling, 2013; Rezende et al., 2014), generative adversarial network (GAN) (Goodfellow et al., 2014; Radford et al., 2016; Arjovsky et al., 2017), normalizing flow (Rezende & Mohamed, 2015; Dinh et al., 2015; 2017; Kingma & Dhariwal, 2018), and autoregressive models (van den Oord et al., 2016b;a; 2017). However, it is non-trivial to train a deep generative model that can converge to a proper minimum of the associated optimization. For example, GAN suffers non-stability, mode collapse, and generative distortion during training. Many insightful algorithms have been proposed to circumvent those issues, including feature engineering (Salimans et al., 2016), various discrimination metrics (Mao et al., 2016; Arjovsky et al., 2017; Berthelot et al., 2017), distinctive gradient penalties (Gulrajani et al., 2017; Mescheder et al., 2018), spectral normalization to discriminator (Miyato et al., 2018), and orthogonal regularization to generator (Brock et al., 2019). What is particularly of interest is that the breakthrough for GANs has been made with a simple technique of progressively growing neural networks of generators and discriminators from low-resolution images to high-resolution counterparts (Karras et al., 2018a). This kind of progressive growing also helps push the state of the arts to a new level by enabling StyleGAN to produce photo-realistic and detail-sharp results (Karras et al., 2018b), shedding new light on wide applications of GANs in solving real problems. This idea of progressive learning is actually a general manner of cognition process (Elman, 1993; Oudeyer et al., 2007), which has been formally named curriculum learning in machine learning (Bengio et al., 2009). The central topic of this paper is to explore a new curriculum for training deep generative models.

To facilitate robust training of deep generative models with noisy data, we propose curriculum learning with clustering. The key contributions are listed as follows:

- We first summarize four representative curricula for generative models, i.e. architecture (generation capacity), semantics (data content), dimension (data space), and cluster (data structure). Among these curricula, cluster curriculum is newly proposed in this paper.

- Cluster curriculum is to treat data according to the centrality of each data point, which is pictorially illustrated and explained in detail. To foster large-scale learning, we devise the active set algorithm that only needs an active data subset of small fixed size for training.
- The geometric principle is formulated to analyze hardness of noisy data and advantage of cluster curriculum. The geometry pertains to counting a small sphere packed in an ellipsoid, on which is based the percolation theory we use.

The research on curriculum learning is diverse. Our work focuses on curricula that are closely related to data attributes, beyond which is not the scope we concern in this paper.

## 2 CURRICULUM LEARNING

Curriculum learning has been a basic learning approach to promoting performance of algorithms in machine learning. We quote the original words from the seminal paper (Bengio et al., 2009) as its definition:

**Curriculum learning.** *"The basic idea is to start small, learn easier aspects of the task or easier sub-tasks, and then gradually increase the difficulty level"* according to pre-defined or self-learned *curricula.*

From cognitive perspective, curriculum learning is common for human and animal learning when they interact with environments (Elman, 1993), which is the reason why it is natural as a learning rule for machine intelligence. The learning process of cognitive development is *gradual* and *progressive* (Oudeyer et al., 2007). In practice, the design of curricula is task-dependent and data-dependent. Here we summarize the representative curricula that are developed for generative models.

**Architecture curriculum**. The deep neural architecture itself can be viewed as a curriculum from the viewpoint of learning concepts (Hinton & Salakhutdinov, 2006; Bengio et al., 2006) or disentangling representations (Lee et al., 2011). For example, the different layers decompose distinctive features of objects for recognition (Lee et al., 2011; Zeiler & Fergus, 2014; Zhou et al., 2016) and generation (Bau et al., 2018). Besides, Progressive growing of neural architectures is successfully exploited in GANs (Karras et al., 2018a; Heljakka et al., 2018; Korkinof et al., 2018; Karras et al., 2018b).

**Semantics curriculum**. The most intuitive content for each datum is the semantic information that the datum conveys. The hardness of semantics determines the difficulty of learning knowledge from data. Therefore, the semantics can be a common curriculum. For instance, the environment for a game in deep reinforcement learning (Justesen et al., 2018) and the number sense of learning cognitive concepts with neural networks (Zou & McClelland, 2013) can be such curricula.

**Dimension curriculum**. The high dimension usually poses the difficulty of machine learning due to the curse of dimensionality (Donoho, 2000), in the sense that the amount of data points for learning grows exponentially with dimension of variables (Vershynin, 2018). Therefore, the algorithms are expected to be beneficial from growing dimensions. The effectiveness of dimension curriculum is evident from recent progress on deep generative models, such as ProGANs (Karras et al., 2018a;b) by gradually enlarging image resolution and language generation from short sequences to long sequences of more complexity (Rajeswar et al., 2017; Press et al., 2017).

## 3 CLUSTER CURRICULUM

For fitting distributions, dense data points are generally easier to handle than sparse data or outliers. To train generative models robustly, therefore, it is plausible to raise cluster curriculum, meaning that generative algorithms first learn from data points close to cluster centers and then with more data progressively approaching cluster boundaries. Thus the stream of feeding data points to models for curriculum learning is the process of clustering data points according to cluster centrality that will be explained in section 3.2. The toy example in Figure 1 illustrates how to form cluster curriculum.

### 3.1 WHY CLUSTERS MATTER

The importance of clusters for data points is actually obvious from geometric point of view. The data sparsity in high-dimensional spaces causes the difficulty of fitting the underlying distribution of

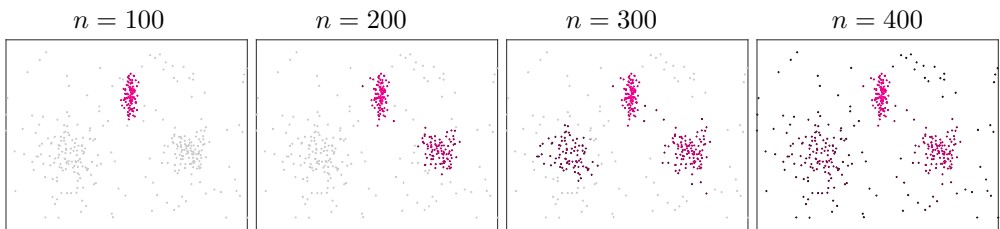

Figure 1: Cluster Curriculum. From magenta color to black color, the centrality of data points reduces. The value $n$ is the number of data points taken with centrality order.

data points (Vershynin, 2018). So generative algorithms may be beneficial when proceeding from the local spaces where data points are relatively dense. Such data points form clusters that are generally informative subsets with respect to the entire dataset. In addition, clusters contain common regular patterns of data points, where generative models are easier to converge. What is most important is that *noisy data points deteriorate performance of algorithms*. For classification, the effectiveness of curriculum learning is theoretically proven to circumvent the negative influence of noisy data (Gong et al., 2016). We will analyze this aspect for generative models with geometric facts.

### 3.2 GENERATIVE MODELS WITH CLUSTER CURRICULUM

With cluster curriculum, we are allowed to gradually learn generative models from dense clusters to cluster boundaries and finally to all data points. In this way, generative algorithms are capable of avoiding the direct harm of noise or outliers. To this end, we first need a measure called centrality that is the terminology in graph-based clustering. It quantifies the compactness of a cluster in data points or a community in complex networks (Newman, 2010). A large centrality implies that the associated data point is close to one of cluster centers. For easy reference, we provide the algorithm of the centrality we use in Appendix. For experiments in this paper, all the cluster curricula are constructed by the centrality of stationary probability distribution, i.e. the eigenvector corresponding to the largest eigenvalue of the transition probability matrix drawn from the data.

To be specific, let $c \in \mathcal{R}^m$ denote the centrality vector of $m$ data points. Namely, the $i$-th entry $c_i$ of $c$ is the centrality of data point $x_i$. Sorting $c$ in descending order and adjusting the order of original data points accordingly give data points arranged by cluster centrality. Let $\overrightarrow{\mathcal{X}} = \{\overrightarrow{\mathcal{X}}_0, \overrightarrow{\mathcal{X}}_1, \ldots, \overrightarrow{\mathcal{X}}_l\}$ signify the set of centrality-sorted data points, where $\overrightarrow{\mathcal{X}}_0$ is the base set that contains sufficient data to attain convergent generative models, and the rest of $\overrightarrow{\mathcal{X}}$ is evenly divided into $l$ subsets according to centrality order. In general, the number of data points in $\overrightarrow{\mathcal{X}}_0$ is much less than $m$ and determined according to $\mathcal{X}$. Such division of $\overrightarrow{\mathcal{X}}_0$ serves to efficiency of training, because we do not need to train models from a very small dataset. The cluster curriculum learning is carried out by incrementally feeding subsets in $\overrightarrow{\mathcal{X}}$ into generative algorithms. In other words, algorithms are successively trained on $\overrightarrow{\mathcal{X}}_0 \leftarrow \overrightarrow{\mathcal{X}}_0 \cup \overrightarrow{\mathcal{X}}_i$ after $\overrightarrow{\mathcal{X}}_0$, meaning that the curriculum for each round of training is accumulated with $\overrightarrow{\mathcal{X}}_i$.

In order to determine the optimal curriculum $\overrightarrow{\mathcal{X}}_0 \cup \overrightarrow{\mathcal{X}}_1 \cup \cdots \cup \overrightarrow{\mathcal{X}}_i$, we need the aid of quality metric of generative models, such as Fréchet inception distance (FID) or sliced Wasserstein distance (SWD) (Borji, 2018). For generative models trained with each curriculum, we calculate the associated score $s_i$ via the specified quality metric. The optimal curriculum for effective training can be identified by the minimal value for all $s_i$, where $i = 1, \ldots, l + 1$. The interesting phenomenon of this score curve will be illustrated in the experiment. The minimum of score $s$ is apparently metric-dependent. One can refer to (Borji, 2018) for the review of evaluation metrics. In practice, we can opt one of reliable metrics to use or multiple metrics for decision-making of the optimal model.

There are two ways of using the incremental subset $\overrightarrow{\mathcal{X}}_i$ during training. One is that the parameters of models are re-randomized when the new data are used, the procedure of which is given in Algorithm 1 in Appendix. The other is that the parameters are fine-tuned based on pre-training of the previous model, which will be presented with a fast learning algorithm in the following section.

### 3.3 ACTIVE SET FOR SCALABLE TRAINING

To obtain the precise minimum of $s$, the cardinality of $\overrightarrow{\mathcal{X}}_i$ needs to be set much smaller than $m$, meaning that $l$ will be large even for a dataset of moderate scale. The training of many loops will be time-consuming. Here we propose the active set to address the issue, in the sense that for each loop of cluster curriculum, the generative models are always trained with a subset of a small fixed size instead of $\overrightarrow{\mathcal{X}}_0 \leftarrow \overrightarrow{\mathcal{X}}_0 \cup \overrightarrow{\mathcal{X}}_i$ whose size becomes incrementally large.

To form the active set $\mathcal{A}$, the subset $\overrightarrow{\mathcal{A}}_0$ of data points are randomly sampled from $\overrightarrow{\mathcal{X}}_0$ to combine with $\overrightarrow{\mathcal{X}}_i$ for the next loop, where $|\overrightarrow{\mathcal{A}}_0| = |\mathcal{A}| - |\overrightarrow{\mathcal{X}}_i|$. For easy understanding, we illustrate the active set with toy example in Figure 2. In this scenario, *progressive pre-training* must be applied, meaning that the update of model parameters for the current training is based on parameters of previous loop. The procedure of cluster curriculum with active set is detailed in Algorithm 2 in Appendix.

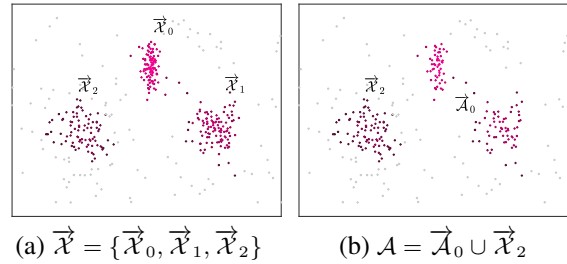

(a) $\overrightarrow{\mathcal{X}} = \{\overrightarrow{\mathcal{X}}_0, \overrightarrow{\mathcal{X}}_1, \overrightarrow{\mathcal{X}}_2\}$     (b) $\mathcal{A} = \overrightarrow{\mathcal{A}}_0 \cup \overrightarrow{\mathcal{X}}_2$

Figure 2: Schematic illustration of active set for cluster curriculum. Here $|\overrightarrow{\mathcal{X}}_0| = |\overrightarrow{\mathcal{X}}_1| = |\overrightarrow{\mathcal{X}}_2| = 100$. The cardinality $|\mathcal{A}|$ of the active set $\mathcal{A}$ is 200. When $\overrightarrow{\mathcal{X}}_2$ is taken for training, we need to randomly sample another 100 (i.e. $|\mathcal{A}| - |\overrightarrow{\mathcal{X}}_2|$) data points from the history data $\overrightarrow{\mathcal{X}}_0 \cup \overrightarrow{\mathcal{X}}_1$ to form $\overrightarrow{\mathcal{A}}_0$. Then the complete active set is composed by $\mathcal{A} = \overrightarrow{\mathcal{A}}_0 \cup \overrightarrow{\mathcal{X}}_2$. We can see that data points in $\overrightarrow{\mathcal{X}}_0 \cup \overrightarrow{\mathcal{X}}_1$ become less dense after sampling.

The active set allows us to train generative models with a small dataset that is actively adapted, thereby significantly reducing the training time for large-scale data.

## 4 GEOMETRIC VIEW OF CLUSTER CURRICULUM

Cluster curriculum bears the interesting relation to high-dimensional geometry, which can provide geometric understanding of our algorithm. Without loss of generality, we work on a cluster obeying the normal distribution. The characteristic of the cluster can be extended into other clusters of the same distribution. For easy analysis, let us begin with a toy example. As Figure 3(a) shows, the confidence ellipse $E_2$ fitted from the subset of centrality-ranked data points is nearly conformal to $E_1$ of all data points, which allows us to put the relation of these two ellipses by virtue of the confidence-level equation. Let $\mathcal{N}(\mathbf{0}, \boldsymbol{\Sigma})$ signify the center and covariance matrix of the cluster $\mathcal{C}$ of interest, where $\mathcal{C} = \{\boldsymbol{x}_i | \boldsymbol{x}_i \in \mathbb{R}^d, i = 1, \dots, n\}$. To make it formal, we can write the equation by

$$\boldsymbol{x}^\top \boldsymbol{\Sigma}^{-1} \boldsymbol{x} = \chi^2_{d,\alpha}, \qquad (1)$$

where $\chi^2_{d,\alpha}$ can be the chi-squared distribution or Mahalanobis distance square, $d$ is the degree of freedom, and $(1 - \alpha) \in [0, 1]$ is the confidence level. For conciseness, we write $\chi^2_{d,\alpha}$ as $\chi^2_\alpha$ in the following context. Then the ellipses $E_{\alpha_1}$ and $E_{\alpha_2}$ correspond to $\chi^2_{\alpha_1}$ and $\chi^2_{\alpha_2}$, respectively, where $\alpha_1 \leq \alpha_2$.

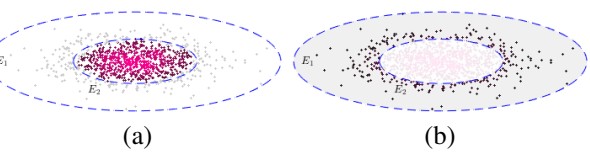

(a)          (b)

Figure 3: Illustration of growing one cluster in cluster curriculum. (a) Data points taken with large centrality. (b) The annulus formed by of removing the inner ellipse from the outer one.

To analyze the hardness of training generative models, a fundamental aspect is to *examine the number $n(E)$ of given data points falling in a geometric entity $E$* [1] *and the number $N(E)$ of lattice points in it*. The less $n(E)$ is compared to $N(E)$, the harder the problem will be. However, the enumeration of lattice points is computationally prohibitive for high dimensions. Inspired by the information theory of encoding data of normal distributions (Roman, 1996; Ma et al., 2007), we count the number of small spheres $S_\varepsilon$ of radius $\varepsilon$ packed in the ellipsoid $E$ instead. Thus we can use this number to replace the role of $N(E)$ as long as the radius of the sphere $S_\varepsilon$ is set properly. With a little abuse of

---

[1]For cluster curriculum, it is an annulus explained shortly.

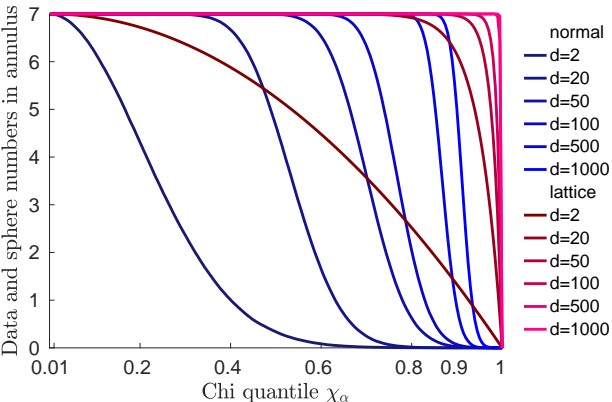

Figure 4: Comparison between the number $n(A)$ of data points sampled from the isotropic normal distributions and $N(A)$ of spheres (lattice) packed in the annulus $A$ with respect to the Chi quantile $\chi_{\alpha_2}$. $d$ is the dimension of data points. For each dimension, we sample 70,000 data points from $\mathcal{N}(\mathbf{0}, \mathbf{I})$. The scales of $y$-axis and $x$-axis are normalized by 10,000 and $\chi_{\alpha_1}$, respectively.

notation, we still use $N(E)$ to denote the packing number in the following context. Theorem 1 gives the exact form of $N(E)$.

**Theorem 1.** *For a set $\mathcal{C} = \{x_i | x_i \in \mathbb{R}^d\}$ of $n$ data points drawn from normal distribution $\mathcal{N}(\mathbf{0}, \mathbf{\Sigma})$, the ellipsoid $E_\alpha$ of confidence $1 - \alpha$ is defined as $x^\top \mathbf{\Sigma}^{-1} x \leq \chi_\alpha^2$, where $\mathbf{\Sigma}$ has no zero eigenvalues and $\alpha \in [0, 1]$. Let $N(E_\alpha)$ be the number of spheres of radius $\varepsilon$ packed in the ellipsoid $E_\alpha$. Then we can establish*

$$N(E_\alpha) = \left(\frac{\chi_\alpha}{\varepsilon}\right)^d \sqrt{\det(\mathbf{\Sigma})}. \tag{2}$$

We can see that $N(E_\alpha)$ admits a tidy form with Mahalanobis distance $\chi_\alpha$, dimension $d$, and sphere radius $\varepsilon$ as variables. The proof is provided in Appendix.

The geometric region of interest for cluster curriculum is the annulus $A$ formed by removing the ellipsoid [2] $E_{\alpha_2}$ from the ellipsoid $E_{\alpha_1}$, as Figure 3(b) displays. We investigate the varying law between $n(A)$ and $N(A)$ in the annulus $A$ when the inner ellipse $E_{\alpha_2}$ grows with cluster curriculum. For this purpose, we need the following two corollaries that immediately follows from Theorem 1.

**Corollary 1.** *Let $N(A)$ be the number of spheres of radius $\varepsilon$ packed in the annulus $A$ that is formed by removing the ellipsoid $E_{\alpha_1}$ from the ellipsoid $E_{\alpha_1}$, where $\alpha_1 \leq \alpha_2$. Then we have*

$$N(A) = \left(\left(\frac{\chi_{\alpha_1}}{\varepsilon}\right)^d - \left(\frac{\chi_{\alpha_2}}{\varepsilon}\right)^d\right)\sqrt{\det(\Sigma)}. \tag{3}$$

**Corollary 2.** $N(A)/N(E_{\alpha_1}) = 1 - \left(\chi_{\alpha_2}/\chi_{\alpha_1}\right)^d.$

It is obvious that $N(A)$ goes infinite when $d \to \infty$ under the conditions that $\chi_{\alpha_1} > \chi_{\alpha_2}$ and $\varepsilon$ is bounded. Besides, when $E_{\alpha_2}$ (cluster) grows, $N(A)$ reduces with exponent $d$ if $E_{\alpha_1}$ is fixed.

In light of Corollary 1, we can now demonstrate the functional law between $n(A)$ and $N(A)$. First, we determine $\chi_{\alpha_1}$ as follows

$$\chi_{\alpha_1} = \max \|x_i\|, \quad x_i \in \mathcal{C}, \tag{4}$$

which means that $E_{\alpha_1}$ is the ellipsoid of minimal Mahalanobis distance to the center that contains all the data points in the cluster. In addition, we need to estimate a suitable sphere radius $\varepsilon$, such that $n(E_{\alpha_1})$ and $N(E_{\alpha_1})$ have comparable scales in order to make $n(A)$ and $N(A)$ comparable in scale. To achieve this, we define an oracle ellipse $E$ where $n(E) = N(E)$. For simplicity, we let $E_{\alpha_1}$ be the oracle ellipse. Thus we can determine $\varepsilon$ with Corollary 3.

**Corollary 3.** *If we let $E_{\alpha_1}$ be the oracle ellipse such that $n(E_{\alpha_1}) = N(E_{\alpha_1})$, then the free parameter $\varepsilon$ can be computed with $\varepsilon = \chi_{\alpha_1}\left(\sqrt{\det(\mathbf{\Sigma})}/n(E_{\alpha_1})\right)^{\frac{1}{d}}.$*

To make the demonstration amenable to handle, data points we use for simulation are assumed to obey the isotropic normal distribution, meaning that data points are generated with nearly equal

---
[2]The ellipse refers to the surface and the ellipsoid refers to the elliptic ball.

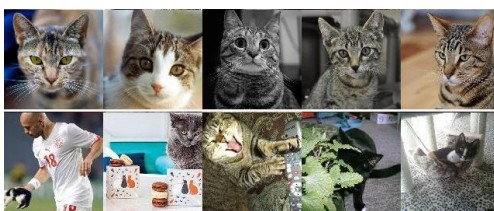 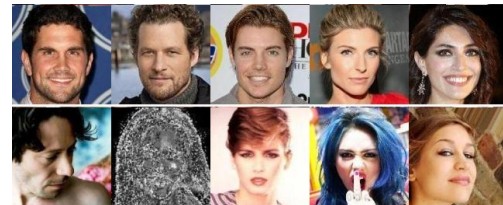

Figure 5: Examples of LSUN cat dataset and CelebA face dataset. The samples in the first row are of high centrality and the samples of low centrality in the second row are noisy data or outliers that we call in the context.

variance along each dimension. Figure 4 shows that $n(A)$ gradually exhibits the critical phenomena of percolation processes[3] when the dimension $d$ goes large, implying that the data points in the annulus $A$ are significantly reduced when $E_{\alpha_2}$ grows a little bigger near the critical point. In contrast, the number $N(A)$ of lattice points is still large and varies negligibly until $E_{\alpha_2}$ approaches the boundary. This discrepancy indicates clearly that fitting data points in the annulus is pretty hard and guaranteeing the precision is nearly impossible when crossing the critical point of $n(A)$ even for a moderate dimension (e.g. $d = 500$). Therefore, the plausibility of cluster curriculum can be drawn naturally from this geometric fact.

## 5 EXPERIMENT

The generative model that we use for experiments are Progressive growing of GAN (ProGAN) (Karras et al., 2018a). This algorithm is chosen because ProGAN is the state-of-the-arts algorithm of GANs with official open sources available. According to convention, we opt the Fréchet inception distance (FID) (Borji, 2018) for ProGAN as the quality metric.

### 5.1 DATASET AND EXPERIMENTAL SETTING

We randomly sample the 200,000 cat images from the LSUN dataset (Yu et al., 2015). These cat images are captured in the wild. So their styles vary significantly. Figure 5 shows the cat examples of high and low centralities. We can see that the noisy cat images differ much from the clean ones. There actually contain the images of very few informative cat features, which are the outliers we refer to. The curriculum parameters are set as $|\vec{\mathcal{X}}_0| = 20,000$ and $|\vec{\mathcal{X}}_i| = 10,000$, which means that the algorithms are trained with 20,000 images first and after the initial training, another 10,000 images according to centrality order are merged into the current training data for further re-training. For active set, its size is fixed to be $30,000$.

The CelebA dataset is a large-scale face attribute dataset (Liu et al., 2015). We use the cropped and well-aligned faces with a bit of image backgrounds preserved for generation task. For cluster-curriculum learning, we randomly sample 70,000 faces as the training set. The face examples of different centralities are shown in Figure 5. The curriculum parameters are set as $|\vec{\mathcal{X}}_0| = 10,000$ and $|\vec{\mathcal{X}}_i| = 5,000$. We bypass the experiment of the active set on faces because it is used for the large-scale data.

Each image in two databases is resized to be $64 \times 64$. To form cluster curricula, we exploit ResNet34 (He et al., 2016) pre-trained on ImageNet (Russakovsky et al., 2015) to extract 512-dimensional features for each face and cat images. The directed graphs are built with these feature vectors. We determine the parameter $\sigma$ of edge weights by enforcing the geometric mean of weights to be 0.8. The robustness of varying the value was validated in (Zhao & Tang, 2008) for clustering. The number of nearest neighbors is set to be $K = 4 \log m$. The centrality is the stationary probability distribution. All codes are written with TensorFlow.

---

[3]Percolation theory is a fundamental tool of studying the structure of complex systems in statistical physics and mathematics. The critical point is the percolation threshold where the transition takes place. One can refer to (Stauffer & Aharony, 1994) if interested.

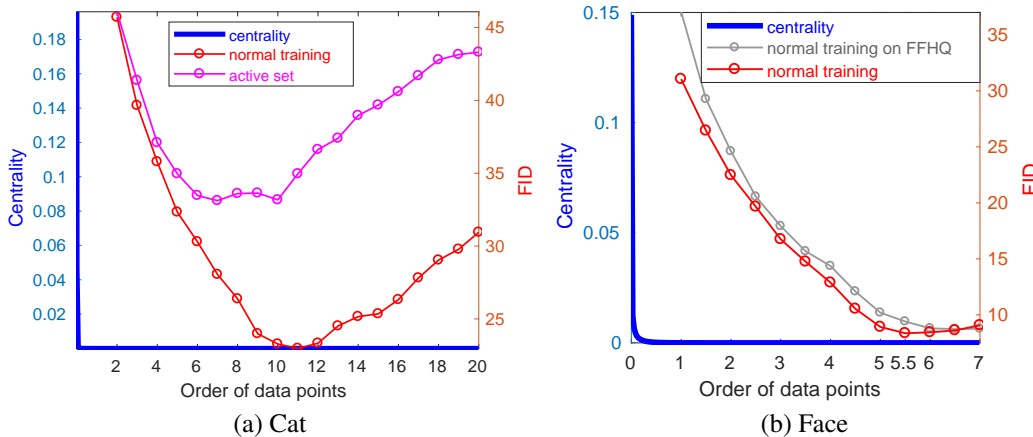

(a) Cat                                                (b) Face

Figure 6: FID curves of cluster-curriculum learning for ProGAN on the cat dataset and CelebA face dataset. The centrality and the FID share the $x$-axis due to that they have the same order of data points. The same colors of the $y$-axis labels and the curves denote the figurative correspondence. The network parameters for "normal training" are randomly re-initialized for each re-training. The active set is based on progressive pre-training of the fixed small dataset. The scale of the $x$-axis is normalized by 10,000.

## 5.2 EXPERIMENTAL RESULT

From Figure 6a, we can see that the FID curves are all nearly V-shaped, indicating that the global minima exist amid the training process. This is the clear evidence that the noisy data and outliers deteriorate the quality of generative models during training. From the optimal curricula found by two algorithms (i.e. curricula at 110,000 and 100,000), we can see that the curriculum of the active set differs from that of normal training only by one-step data increment, implying that the active set is reliable for fast cluster-curriculum learning. The performance of the active set measured by FID is much worse than that of normal training, especially when more noisy data are fed into generative models. However, this does not change the whole V-shape of the accuracy curve. Namely, it is applicable as long as the active set admits the metric minimum corresponding to the appropriate curriculum.

The V-shape of the centrality-FID curve on the cat data is due to that the noisy data of low centrality contains little effective information to characterize the cats, as already displayed in Figure 5. However, it is different for the CelebA face dataset where the face images of low centrality also convey the part of face features. As evident by Figure 6b, ProGAN keeps being optimized by the majority of the data until the curriculum of size $55,000$. To highlight the meaning of this nearly negligible minimum, we also conduct the exactly same experiment on the FFHQ face dataset containing $70,000$ face images of high-quality (Karras et al., 2018b). For FFHQ data, the noisy face data can be ignored. The gray curve of normal training in Figure 6b indicates that the FID of ProGAN is monotonically decreased for all curricula. This gentle difference of the FID curves at the ends between CelebA and FFHQ clearly demonstrates the difficulty of noisy data to generative algorithms.

## 5.3 GEOMETRIC INVESTIGATION

To understand cluster curriculum deeply, we employ the geometric method formulated in section 4 to analyze the cat and face data. The percolation processes are both conducted with 512-dimensional features from ResNet34. Figure 7 displays the curve of $n(A)$ that is the variable of interest in this scenario. As expected, the critical point in the percolation process occurs for both cases, as shown by blue curves. An obvious fact is that the optimal curricula (red strips) both fall into the (feasible) domains of percolation processes after the critical points, as indicated by gray color. This is a desirable property because data become rather sparse in the annuli when crossing the critical points. Then noisy data play the non-negligible role on tuning the parameters of generative models. Therefore, a fast learning strategy can be derived from the percolation process. Training may begin from the curriculum specified by the critical point, thus significantly accelerating cluster-curriculum learning.

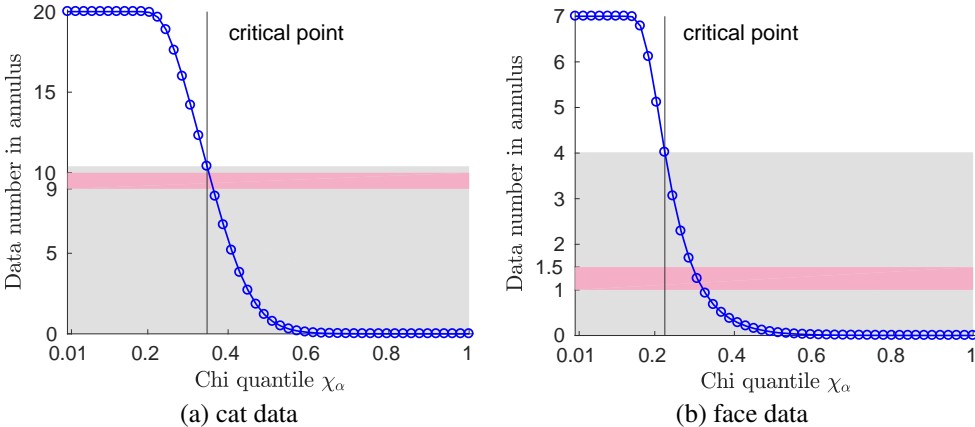

(a) cat data                    (b) face data

Figure 7: Geometric phenomenon of cluster curriculum on the LSUN cat and CelebA face datasets. The pink strips are intervals of optimal curricula derived by generative models. For example, the value 9 of the pink interval in (a) is obtained by $9 = 20 - 11$, where 11 is one of the minima (i.e. 110,000) in Figure 6a. The others are derived in the same way. The subtraction transforms the data number in the cluster to be the one in the annulus. The critical points are determined by searching the maxima of the absolute discrete difference of the associated curves. The scales of $y$-axes are normalized by 10,000.

Another intriguing phenomenon is that the more noisy the data, the closer the optimal interval (red strip) is to the critical point. We can see that the optimal interval of the cat data is much closer to the critical point than that of the face data. What surprises us here is that the optimal interval of cluster curricula associated with the cat data nearly coincides with the critical point of the percolation process in the annulus! This means that the optimal curriculum may be found at the intervals close to the critical point of of $n(A)$ percolation for heavily noisy data, thus affording great convenience to learning an appropriate generative model for such datasets.

## 6    ANALYSIS AND CONCLUSION

Cluster curriculum is proposed for robust training of generative models. The active set of cluster curriculum is devised to facilitate scalable learning. The geometric principle behind cluster curriculum is analyzed in detail as well. The experimental results on the LSUN cat dataset and CelebA face dataset demonstrate that the generative models trained with cluster curriculum is capable of learning the optimal parameters with respect to the specified quality metric such as Fréchet inception distance and sliced Wasserstein distance. Geometric analysis indicates that the optimal curricula obtained from generative models are closely related to the critical points of the associated percolation processes established in this paper. This intriguing geometric phenomenon is worth being explored deeply in terms of the theoretical connection between generative models and high-dimensional geometry.

It is worth emphasizing that the meaning of model optimality refers to the global minimum of the centrality-FID curve. As we already noted, the optimality is metric-dependent. We are able to obtain the optimal model with cluster curriculum, which does not mean that the algorithm only serves to this purpose. We know that more informative data can help learn a more powerful model covering the large data diversity. Here a trade-off arises, i.e. the robustness against noise and the capacity of fitting more data. The centrality-FID curve provides a visual tool to monitor the state of model training, thus aiding us in understanding the learning process and selecting suitable models according to noisy degree of given data. For instance, we can pick the trained model close to the optimal curriculum for heavily noisy data or the one near the end of the centrality-FID curve for datasets of little noise. In fact, this may be the most common way of using cluster curriculum.

In this paper, we do not investigate the cluster-curriculum learning for the multi-class case, e.g. the ImageNet dataset with BigGAN (Brock et al., 2019). The cluster-curriculum learning of multiple classes is more complex than that we have already analyzed on the face and cat data. We leave this study for future work.

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

## A  APPENDIX

### A.1  CENTRALITY MEASURE

The centrality or clustering coefficient pertaining to a cluster in data points or a community in a complex network is a well-studied traditional topic in machine learning and complex systems. Here we introduce the graph-theoretic centrality for the utilization of cluster curriculum. Firstly, we construct a directed graph (digraph) with $K$ nearest neighbors by the method in (Zhao & Tang, 2008). The weighted adjacency matrix $\boldsymbol{W}$ of the digraph can be formed in this way: $W_{ij} = \exp(-d_{ij}^2/\sigma^2)$ if $\boldsymbol{x}_j$ is one of the nearest neighbors of $\boldsymbol{x}_i$ and 0 otherwise, where $d_{ij}$ is the distance between $\boldsymbol{x}_i$ and $\boldsymbol{x}_j$ and $\sigma$ is a free parameter.

The density of data points can be quantified with the stationary probability distribution of a Markov chain. For a digraph built from data, the transition probability matrix can be derived by row normalization, say, $P_{ij} = W_{ij}/\sum_j W_{ij}$. Then the stationary probability $\boldsymbol{u}$ can be obtained by solving an eigenvalue problem

$$\boldsymbol{P}^\top \boldsymbol{u} = \boldsymbol{u}, \tag{5}$$

where $\top$ denotes the matrix transpose. It is straightforward to know that $\boldsymbol{u}$ is the eigen-vector of $\boldsymbol{P}^\top$ corresponding to the largest eigenvalue (i.e. 1). $\boldsymbol{u}$ is also defined as a kind of PageRank in many scenarios.

For density-based cluster curriculum, the centrality $\boldsymbol{c}$ coincides with the stationary probability $\boldsymbol{u}$. Figure 1 in the main context shows the plausibility of using the stationary probability distribution to quantify the data density.

## A.2 Theorem 1 and Proof

**Theorem 1.** *For a set $\mathcal{C} = \{x_i | x_i \in \mathbb{R}^d\}$ of $n$ data points drawn from normal distribution $\mathcal{N}(\mathbf{0}, \mathbf{\Sigma})$, the ellipsoid $E_\alpha$ of confidence $1 - \alpha$ is defined as $x^\top \mathbf{\Sigma}^{-1} x \leq \chi_\alpha^2$, where $\mathbf{\Sigma}$ has no zero eigenvalues and $\alpha \in [0, 1]$. Let $N(E_\alpha)$ be the number of spheres of radius $\varepsilon$ packed in the ellipsoid $E_\alpha$. Then we can establish*

$$N(E_\alpha) = \left(\frac{\chi_\alpha}{\varepsilon}\right)^d \sqrt{\det(\mathbf{\Sigma})}. \tag{6}$$

*Proof.* As explained in the main context, the ellipse equation with respect to the confidence $\alpha$ can be expressed by the following equation

$$x^\top \mathbf{\Sigma}^{-1} x = \chi_\alpha^2. \tag{7}$$

Suppose that $\lambda_1, \ldots, \lambda_d$ are the eigenvalues of $\mathbf{\Sigma}$. Then equation (7) can be written as

$$\frac{x_1^2}{\lambda_1} + \cdots + \frac{x_d^2}{\lambda_d} = \chi_\alpha^2. \tag{8}$$

Further, eliminating $\chi_\alpha^2$ on the right side gives

$$\frac{x_1^2}{\chi_\alpha^2 \lambda_1} + \cdots + \frac{x_d^2}{\chi_\alpha^2 \lambda_d} = 1. \tag{9}$$

Then we derive the length of semi-axis with respect to $\chi_\alpha$, i.e.

$$r_i = \chi_\alpha \sqrt{\lambda_i}. \tag{10}$$

For a $d$-dimensional ellipsoid $E$, the volume of $E$ is

$$\text{Vol}(E_\alpha) = \frac{2}{d} \frac{\pi^{d/2}}{\Gamma(d/2)} r_1 r_2 \cdots r_d, \tag{11}$$

where $r_i$ the leng of semi-axis of $E$ and $\Gamma(\cdot)$ is the Gamma function. Substituting (10) into the above equation, we obtain the final formula of volume

$$\text{Vol}(E_\alpha) = \frac{2}{d} \frac{\pi^{d/2}}{\Gamma(d/2)} \chi_\alpha^d \prod_{i=1}^{d} \sqrt{\lambda_i}, \tag{12}$$

$$= \frac{2}{d} \frac{\pi^{d/2}}{\Gamma(d/2)} \chi_\alpha^d \sqrt{\det(\mathbf{\Sigma})}. \tag{13}$$

Using the volume formula in (11), it is straightforward to get the volume of the packing sphere $S_\varepsilon$

$$\text{Vol}(S_z) = \frac{2}{d} \frac{\pi^{d/2}}{\Gamma(d/2)} \sqrt{\det(\varepsilon^2 \boldsymbol{I})}. \tag{14}$$

By the definition of $N(E_\alpha)$, we can write

$$N(E_\alpha) = \text{\# of spheres} = \text{Vol}(E_\alpha)/\text{Vol}(S_\varepsilon) \tag{15}$$

$$= \left(\frac{\chi_\alpha}{\varepsilon}\right)^d \sqrt{\det(\mathbf{\Sigma})}. \tag{16}$$

We conclude the proof of the theorem. $\qquad\square$

## A.3 PROCEDURES OF ALGORITHM 1 AND ALGORITHM 2

---

**Algorithm 1** Cluster Curriculum for Generative Models

---

**Require:**
1: $\mathcal{X}$, dataset containing $m$ data points
2: `GenerativeModel()`, generative models
3: `QualityScore()`, metric for generative results
4: $l$, number of subsets
**Ensure:**
5: // Solve centralities
6: $c = $ `Centrality`$(\mathcal{X})$           $\triangleright$ section A.1
7: // Cluster curriculum
8: Get sorted data $\overrightarrow{\mathcal{X}}$ according to descending order of $c$
9: Divide $\overrightarrow{\mathcal{X}}$ to be $\{\overrightarrow{\mathcal{X}}_0, \overrightarrow{\mathcal{X}}_1, \ldots, \overrightarrow{\mathcal{X}}_l\}$
10: // Train generative models
11: **for** $\overrightarrow{\mathcal{X}}_i \in \{\overrightarrow{\mathcal{X}}_0, \overrightarrow{\mathcal{X}}_1, \ldots, \overrightarrow{\mathcal{X}}_l\}$ **do**
12:    Initialize model parameters $\theta$ randomly
13:    $\theta_i = $ `GenerativeModel`$(\overrightarrow{\mathcal{X}}_0, \theta)$        $\triangleright$ e.g. GAN
14:    $\overrightarrow{\mathcal{X}}_0 \leftarrow \{\overrightarrow{\mathcal{X}}_0 \cup \overrightarrow{\mathcal{X}}_i\}$
15: **end for**
16: // Search the optimal set
17: **for** $\theta_i \in \{\theta_1 \ldots, \theta_{l+1}\}$ **do**
18:    Generate data $\overline{\mathcal{X}}$ with model of parameter $\theta_i$
19:    $s_i = $ `QualityScore`$(\overline{\mathcal{X}})$          $\triangleright$ e.g. FID
20: **end for**
21: $i^* = \arg\min s_i, i = 1, \ldots, l + 1$
22: Return the optimal model parameter $\theta_{i^*}$

---

---

**Algorithm 2** Cluster Curriculum with Active Set

---

**Require:**
1: $\mathcal{X}$, GenerativeModel(), QualityScore() as in Algorithm 1
2: $|\mathcal{A}|$, cardinality of active set $\mathcal{A}$
**Ensure:**
3: // Solve centralities
4: $\boldsymbol{c} = $ Centrality($\mathcal{X}$) $\qquad\qquad\qquad\qquad\qquad\qquad\qquad\qquad\qquad$ ▷ section A.1
5: // Cluster curriculum
6: Get sorted data $\overrightarrow{\mathcal{X}}$ according to descending order of $\boldsymbol{c}$
7: Divide $\overrightarrow{\mathcal{X}}$ to be $\{\overrightarrow{\mathcal{X}}_0, \overrightarrow{\mathcal{X}}_1, \ldots, \overrightarrow{\mathcal{X}}_l\}$
8: // Train generative models
9: Initialize model parameters $\boldsymbol{\theta}$ randomly
10: $\boldsymbol{\theta}_0 = $ GenerativeModel($\overrightarrow{\mathcal{X}}_0, \boldsymbol{\theta}$) $\qquad\qquad\qquad\qquad\qquad\qquad$ ▷ e.g. GAN
11: **for** $\overrightarrow{\mathcal{X}}_i \in \{\overrightarrow{\mathcal{X}}_1, \ldots, \overrightarrow{\mathcal{X}}_l\}$ **do**
12: $\qquad$ Derive $\overrightarrow{\mathcal{A}}_0$ by randomly sampling $\overrightarrow{\mathcal{X}}_0$
13: $\qquad$ $\overrightarrow{\mathcal{A}}_0 \leftarrow \{\overrightarrow{\mathcal{A}}_0 \cup \overrightarrow{\mathcal{X}}_i\}$
14: $\qquad$ // Use pre-training model
15: $\qquad$ $\boldsymbol{\theta}_i = $ GenerativeModel($\overrightarrow{\mathcal{A}}_0, \boldsymbol{\theta}_{i-1}$)
16: $\qquad$ $\overrightarrow{\mathcal{X}}_0 \leftarrow \{\overrightarrow{\mathcal{X}}_0 \cup \overrightarrow{\mathcal{X}}_i\}$
17: **end for**
18: // Search the optimal set
19: **for** $\boldsymbol{\theta}_i \in \{\theta_1 \ldots, \theta_{l+1}\}$ **do**
20: $\qquad$ Generate data $\overline{\mathcal{X}}$ with model of parameter $\boldsymbol{\theta}_i$ by sampling a prior $\qquad$ ▷ e.g. Gaussian
21: $\qquad$ $s_i = $ QualityScore($\overline{\mathcal{X}}$) $\qquad\qquad\qquad\qquad\qquad\qquad\qquad\qquad$ ▷ e.g. FID
22: **end for**
23: $i^* = \arg\min s_i, i = 1, \ldots, l+1$
24: Return the optimal model parameter $\boldsymbol{\theta}_{i^*}$

---

