# OpenReview forum: "Curriculum Learning for Deep Generative Models with Clustering"
_ICLR.cc/2020/Conference — Reject_

### Official Review · AnonReviewer2 · 2019-10-21
**Official Blind Review #2**

**Rating:** 6

**Review:**

Summary: This well written paper presents an effective way to remove outliers for deep generative models, provided examples are ranked along their centrality. One question I have is how exactly this centrality is measured

The word ‘curriculum’ has become a terminology used to describe a wide variety of totally different algorithms. While the authors provide an excellent introduction to this diversity and clearly differentiate their own flavor of ‘cluster curriculum’, I am wondering if they would not have been better off by describing the proposed approach in terms of outlier removal, especially has it has very little in common with the original idea of curriculum, which is a learning progression designed by the teacher.

Nevertheless, the paper is very clearly written and reads well in the present form. I am especially impressed about the clarity of the rather technical part on percolation.

In terms of outlier removal, this could be interpreted as the following constructive algorithm:
•	Examples are ranked along their centrality measure
•	One identify the critical point in percolation for a starting point where too many examples are removed
•	More examples are added until a minimum is found on validation data
This process can be made faster as:
•	The optimum can happen very soon after the critical point
•	Otherwise, an active set algorithm can be used to control the size of the training set.

While my recent expertise has been more NLP than Vision, I think this algorithm is original, and could have a significant impact as it can be extended beyond GANs. The technical presentation is excellent.

One puzzling issue is the computation of the centrality measure: all I could find in Appendix A1. and A.3 is that it is directly measured on the raw image (RGB pixels?) by taking some distance, which I assume is Euclidean? My experience in image classification suggests such a distance is meaningless, so I may be missing something. I looked for a Github pointer, but none was provided.

The English is OK, but there are missing words and strange constructions:
•	Page 1:  “Deep generative models HAVE piqueD researchers’ interest in the past decade”
•	Page 4: “The training of many loops will lead to time-consuming (MISSING WORD)”

In particular, the usage of “the”, for instance in pages:
•	7 “Therefore, a fast learning strategy can be derived from THE percolation process.”
•	7 “Training may begin from the curriculum”
•	8 “Cluster curriculum is proposed for robust training of generative models.”


**Experience Assessment:**

I have read many papers in this area.

**Review Assessment: Checking Correctness Of Derivations And Theory:**

I assessed the sensibility of the derivations and theory.

**Review Assessment: Checking Correctness Of Experiments:**

I assessed the sensibility of the experiments.

**Review Assessment: Thoroughness In Paper Reading:**

I read the paper at least twice and used my best judgement in assessing the paper.

---

> ### Author Response · Authors · 2019-11-07
> **To Reviewer #2**
>
>
> Q1:  “I am wondering if they would not have been better off by describing the proposed approach in terms of outlier removal.”
> A1: Indeed, outlier removal is more clear and easier to understand. But we use “clustering” to emphasize the dynamic process when using cluster curriculum for training generative models.
>
> Q2:  “One puzzling issue is the computation of the centrality measure.”
> A2: We used ResNet34 to extract 512-dimensional features for each image.  The pre-trained model on ImageNet is available at
>
> ResNet34
> https://github.com/qubvel/classification_models
>
> Then the graph is constructed by Algorithm 3 presented in the following NeurIPS paper
>
> Cyclizing Clusters via Zeta Function of a Graph
> https://papers.nips.cc/paper/3412-cyclizing-clusters-via-zeta-function-of-a-graph
>
> In this paper, the authors validated the effectiveness of using graphs for clustering complex data. We will make this clear in the revised version.
>
> We will polish our paper according to your advice on writing. Thank you for your careful review and insightful comments on our work. We appreciate them very much.

---

### Official Review · AnonReviewer3 · 2019-10-31
**Official Blind Review #3**

**Rating:** 1

**Review:**

The paper states that training generative models is challenging when there are noisy data points in your training set. To address this, the authors propose to training methodology (or curricula)  where "easier" or more "relevant" points are presented first followed to the model followed by the less relevant ones. The relevance is determined used by calculating centrality on a graph constructed out of the data points.

I lean towards rejecting the paper, primarily because 1) The authors have not provided evidence to claim the noisy data points makes training challenging or characterized anything about under how much noise the training breaks down. 2) Experimental evidence is not convincing  3) There are not of imprecise statements in the paper.

The authors claim (without citation) that training generative models in the presence of noisy data is challenging. How much noise are we talking about? If the entire training set is very noisy, maybe we don't have any hope to learn to generate clean samples, but if it's just a little bit of noise, maybe it's fine. I also understand that when the authors use the word "noise" they don't really mean noise,  but changes in view point etc. Some convincing demonstration that such characteristics of dataset adversely affects the training of generative models will be helpful (in addition to one passing sentence in the experiment section about it).

I am not convinced by any of the experiments. More importantly, it seems like the proposed training curricula is in general valid for a lot of generative models. The experimental evidence is really not convincing. Very limited experiments is done on two datasets only using one particular GAN. To convincingly demonstrate that your training methodology is doing something non-trivial, you should show that this works on multiple generative models on multiple datasets and compare your performance (and visualize some generated samples) against just training blindly on the entire dataset. In addition to this, I don't understand the relevance of some of the results in the paper. For ex, what does Corollary 3 signify?

Some other points:
1. First para. "non-trial' -> non-trivial
2. First para: "model collapse" -> mode collapse
3. 3.2 para 2. "base set that guarantees a proper converge of generative models". What do you mean by "proper convergence"?
4. 3.2 para 2. "moderate compared to m". Again, what is moderate?
5. Why do you use ResNet features (5.1) for distance? Why is this a reasonable or the best metric while computing your graph?
6. "We determine the parameter \alpha of the edge weight by enforcing geometric mean of the weights to be 0.8". It seems very arbitrary to me. Can you justify this choice?





**Experience Assessment:**

I do not know much about this area.

**Review Assessment: Checking Correctness Of Derivations And Theory:**

I assessed the sensibility of the derivations and theory.

**Review Assessment: Checking Correctness Of Experiments:**

I assessed the sensibility of the experiments.

**Review Assessment: Thoroughness In Paper Reading:**

I read the paper at least twice and used my best judgement in assessing the paper.

---

> ### Author Response · Authors · 2019-11-07
> **To Reviewer #3**
>
> Q1:  “The authors have not provided evidence to claim the noisy data points makes training challenging or characterized anything about under how much noise the training breaks down.” And all the comments in paragraph 3.
> A1: Please refer to Figure 6(a), where the process is clearly illustrated. We also explain the figure clearly in the context.
>
> Q2: “Experimental evidence is not convincing”,  “Very limited experiments is done on two datasets only using one particular GAN. ” , “you should show that this works on multiple generative models on multiple datasets ”
> A2: Yes, there are multiple generative models. But VAEs and GANs are suitable for high-dimensional data like images. VAE works well for patterns of simple structures like hand-written digits. However, it cannot produce photo-realistic results for complex objects. Therefore, it is less meaningful to use VAEs to perform such experiments.
>
> The ProGAN we used is the oral work published on ICLR 2018 (https://openreview.net/forum?id=Hk99zCeAb). It is the state-of-the-art  GAN  that has accepted 1076 citations (Google citation result up to now) in the past two years. In other words, ProGAN has wide impact on generative models. This is why we chose ProGAN to test our algorithms.
>
> The datasets of cats and faces we used for experiments are benchmark datasets for GANs now. We have already verified the effectiveness of our algorithms on these two complex data.
>
> Q3: “There are not of imprecise statements in the paper.”, “"base set that guarantees a proper converge of generative models". What do you mean by "proper convergence"?”,   “"moderate compared to m". Again, what is moderate? ”
> A3: To make them easily understood, we change the words in the revised version.
>        "base set that guarantees a proper converge of generative models" -> base set that contains sufficient data to attain convergent generative models
>        “moderate compared to m” -> usually less than m
> Thanks for the advice. We will further polish our paper.
>
> Q4: “I don't understand the relevance of some of the results in the paper. For ex, what does Corollary 3 signify?”
> A4: Corollary 3 is to solve epsilon used in equation (3). We explain this clearly in the context.
>
> Q5:  “Why do you use ResNet features (5.1) for distance? Why is this a reasonable or the best metric while computing your graph?”
> A5: Using ResNet to extract features of images is a very common way in computer vision. We follow this convention. But “the best metric” is really hard to be determined, not only for our work but also for nearly all the tasks for feature extraction. It is beyond the scope of our work.
>
> Q6: “ "We determine the parameter \alpha of the edge weight by enforcing geometric mean of the weights to be 0.8". It seems very arbitrary to me. Can you justify this choice?”
> A6: Actually, we determine the parameter by referring to the following NeurIPS paper
>
> Cyclizing Clusters via Zeta Function of a Graph
> https://papers.nips.cc/paper/3412-cyclizing-clusters-via-zeta-function-of-a-graph
>
> where the robustness of choosing the geometric mean is validated for graphs. We will make this clear in the paper.

---

### Author Response · Authors · 2019-11-11
**The revised version (11/11/2019)**


We revised our paper from the following two aspects in this version.

1. We corrected the typos and polished the writing according to Reviewers' advice.

2. We cited the following paper as the reference for the directed graph construction. We used the algorithm presented in this paper for the digraph construction. We made this clear in the revised version.

Cyclizing Clusters via Zeta Function of a Graph
https://papers.nips.cc/paper/3412-cyclizing-clusters-via-zeta-function-of-a-graph

---

### Decision · Program_Chairs · 2019-12-19

**Decision:**

Reject

**Comment:**

The paper proposes a curriculum learning approach to training generative models like GANs. The reviewers had a number of questions and concerns related to specific details in the paper and experimental results. While the authors were able to address some of these concerns, the reviewers believe that further refinement is necessary before the paper is ready for publication.